# Disparate impact pandemic framing decreases public concern for health consequences

**Ugur Yildirim***

Department of Sociology, University of California-Berkeley, Berkeley, CA, United States of America

* ugur.yildirim@berkeley.edu

## Abstract

It is known that the new coronavirus (COVID-19) is disproportionately affecting the elderly, those with underlying medical conditions, and the poor. What is the effect of informing the public about these inequalities on people's perceptions of threat and their sensitivity to the outbreak's human toll? This study answers this question using a novel survey experiment and finds that emphasis on the unequal aspect of the pandemic, especially as it relates to the elderly and those with medical conditions, could be causing the public to become less concerned about the outbreak and its human toll. Discussion situates this finding in the literature on scientific communication and persuasion and explains why language that emphasizes the impact of the virus on *all of us*—rather than singling out certain groups—could be more effective in increasing caution among the general public and make them take the situation more seriously.

## Introduction

Within a few months after its first emergence in Wuhan, China in December 2019, the novel coronavirus (COVID-19) has spread to almost every country on earth, including the US [1]. As of September 2020, the human toll of the disease worldwide is more than 30 million confirmed cases and nearly one million deaths [2]. Very few disease outbreaks in history have had such a fast and widespread impact on humanity, with the closest example being the 1918 flu pandemic [3].

Despite the global nature of the outbreak that has impacted peoples of all sexes, races, and cultural backgrounds, it is known that the disease is not affecting everyone in the same way. In particular, the elderly and those with underlying medical conditions are at higher risk of severe illness due to the virus [4]. Similarly, more infections and deaths are reported in poor and low-income communities compared to wealthier ones [5]. Neither of these patterns are surprising given what we know about health disparities [6–10] and the unequal impact of epidemics on certain groups [11, 12].

While the outbreak is far from having a uniform impact on different groups, the way the media and the scientific community is talking about the outbreak does not always touch upon this unequal aspect of the pandemic. Oftentimes, the account instead emphasizes the

**Data Availability Statement:** All relevant data are within the manuscript and its Supporting Information files.

**Funding:** UY received a small grant from the Experimental Social Science Laboratory (XLAB) at

University of California Berkeley (https://xlab.berkeley.edu/) to run this study. The funders had no role in study design, data collection and analysis, decision to publish, or preparation of the manuscript.

**Competing interests:** The authors have declared that no competing interests exist.

*equalizing* aspect of the pandemic, whereby the virus threatens all of us—all Americans or the entirety of humanity—regardless of our background [13]. Other times, the discussion revolves specifically around how the pandemic has been especially hard on certain groups, such as the elderly and the sick [14].

How do these different framings of the pandemic affect the public opinion? In particular, is one framing more or less effective than the other in terms of how it influences whether or not the public sees the outbreak as a serious threat or not and whether it is more important to save lives or to save the economy as the outbreak unfolds? This study answers this question using a novel survey experiment and finds that emphasis on the unequal aspect of the pandemic, especially as it relates to the elderly and those with medical conditions, could be causing the public to become less concerned about the outbreak and its human toll. Discussion situates this finding in the literature on scientific communication and persuasion and explains why language that emphasizes the impact of the virus on *all of us*—rather than singling out certain groups—could be more effective in increasing caution among the general public and make them take the situation more seriously.

## Materials and methods

The project has IRB approval from University of California-Berkeley (protocol type: Soc-Behav-Ed Exempt; protocol number: 2020-04-13247; protocol title: Perceptions of inequality during the coronavirus outbreak). Written consent was obtained from respondents at the start of the survey.

### Experimental design

The study is designed as a between-subjects survey experiment. It randomized each respondent into one of three conditions corresponding to three possible framings of the pandemic: (1) the "equal pandemic" framing, which does not say anything about the disparate impact of the pandemic on different groups but instead emphasizes how the outbreak has been affecting everyone regardless of their background; (2) the "elderly and medical conditions inequality" framing, which specifically emphasizes the unequal aspect of the pandemic in that it has been especially hard on the elderly and those with medical conditions; and (3) the "class inequality" framing, which specifically emphasizes the unequal aspect of the pandemic in that it has been especially hard on the poor and low-income communities. These conditions are chosen to reflect the ways that the pandemic is discussed in public discourse.

The experiment flows as follows. First, respondents are recruited into the study and asked to give their consent. (At this stage, respondents are told that the goal of the survey is to "understand the public's opinions regarding important societal and economic trends in the US." This general wording is chosen over using specific words such as coronavirus and inequality in an attempt to make sure respondents are not primed to think about these issues from the start.) Second, they are asked to watch a short clip with subtitles and told that the purpose of showing this video is to assess their comprehension skills; the content of the clips depends on the experimental condition respondents are in. Third, right after watching the video, they are asked to briefly describe the content of the video using their own words. Fourth, they answer a series of general questions related to their attitudes towards inequality as well as their socio-demographic characteristics such age, gender, race, and income.

Finally, respondents answer questions that are specifically related to the coronavirus outbreak. These questions include: (1) whether the respondent thinks the coronavirus is a serious threat to the American people or not; (2) whether the respondent thinks it is more important to save lives or to save the economy during this outbreak; how satisfied the respondent is with

the way (3) their city, (4) their state, and (5) the federal government has been handling the coronavirus situation; (6) how the respondent has been affected by the coronavirus outbreak; and (7) how many times the respondent went outside in the past seven days.

Answers given to questions (1) and (2) constitute the main dependent variables in the study. Both variables take values between 1 and 5 with higher values denoting higher threat perceptions in the case of the first variable (1 = not a threat at all, 2 = a small threat, 3 = a threat, 4 = a serious threat, 5 = a very serious threat) and attaching more importance to saving the economy over saving lives in the case of the second variable (1 = saving lives must be the priority even if it means the economy will suffer, 2, 3, 4, 5 = saving the economy must be the priority even if it means lives will be lost). Answers given to questions (3), (4), and (5) are similarly coded to take values between 1 and 5 with higher values denoting more satisfaction.

Multiple binary variables have been generated based on question (6), including whether the respondent or someone in the respondent's family (i) is at risk, (ii) has contracted the virus, (iii) lost their job due to the outbreak, or (iv) experienced a significant decrease in income due to the outbreak. The "at risk" variable is particularly important here because given that the current crisis is caused by a disease outbreak, those who are at risk of severe illness and death will likely view and respond to the crisis very differently compared to those who are not at risk. The variable based on Question (7) takes values between 0 and 7. (See S1 Appendix for the experimental texts, images, videos, manipulation check question, survey questions, and other related project content including additional variables and conditions. The study design is pre-registered, while the specific hypotheses tested in this paper are not.)

## Implementation and subject recruitment

The survey experiment is implemented using Qualtrics. The videos presented to respondents as part of the experiment are prepared using iMovie and subsequently uploaded to a YouTube channel created by the researcher (videos are "unlisted", have comments disabled, and show subtitles by default). All videos showed an Adobe Stock licensed image in the background related to the content of the narrated text. The experimental texts themselves are written by the researcher after a careful reading of relevant news articles and scientific communications.

The texts narrated to respondents in the videos are recorded by a young female in her 20's speaking Standard American English. Female voice is chosen over male voice due to evidence that shows that people tend to find the female voice to be more credible [15]. The narrated text is also displayed as actual text under the video in case the respondent experiences a problem watching the video or chooses not to watch. (As discussed later under Results, the researcher confirmed that most respondents watched and understood the videos.)

Data collection took place on Lucid Theorem. This platform gives researchers access to cheap, fast (thousands of responses within hours), and high quality data that is also nationally representative based on age, gender, ethnicity, and region. A recent scholarly work also validated the quality of Lucid samples [16]. (All code, materials, and de-identified data will be made public once the study is over.)

## Sample characteristics and data structure

The survey experiment is run on a total of 2,617 respondents with approximately 870 respondents in each condition. The three conditions appear to be balanced on the demographic covariates, which gives us confidence that randomization worked as expected. All analyses are conducted on a dataset with the following simple structure: one row per respondent and as many columns as there are variables. Respondents are required to be US residents and 18 or

older. (See S2 Appendix for information on sample size calculations, exact sample sizes by condition, and summary demographics by condition.)

## Overview of statistical models used

Linear regression models are fit to data with the experimental condition as the independent variable. The equal pandemic condition is used as the reference category to be able to get estimates for the elderly and medical conditions inequality and class inequality conditions. (Note that the choice of reference category is somewhat arbitrary as it can be reasonably argued that equal pandemic is actually the distinct frame here. Accordingly, additional models were fit to data—see S3 Appendix—that treat the inequality conditions as the reference category to estimate an equal pandemic effect. These additional models do not change our substantive conclusions at all but allow us to see the story from the opposite angle.)

Since the inclusion of socio-demographic covariates does not change our conclusions—this is not surprising as the independent variable is randomly assigned to respondents—the main text only discusses models without these covariates. (S3 Appendix presents results both with and without socio-demographic covariates for the sake of transparency in line with recent scholarly work [17]. Models with additional outcomes as well as results based on ordinal logistic regression models—which do not change the substantive conclusions discussed in the text—are also presented.)

## Results

### Manipulation checks

Manipulation checks are used in experimental research to determine whether the subjects actually received the treatments the researcher intended them to receive. The researcher confirmed that most respondents actually watched the videos by checking the number of YouTube "views" of each video. Most respondents also passed the manipulation check question, that is, clearly understood the text being communicated to them. (The researcher used a custom script to look for certain keywords such as "coronavirus" or "elderly" to make sure that respondents' description of the video was correct.) Furthermore, conclusions presented here remain unchanged regardless of whether or not we restrict the sample to only those respondents who passed the manipulation check.

### Main findings

The experiment had a significant impact on respondents' opinions regarding whether coronavirus is a serious threat or not and whether the priority should be saving lives or saving the economy. As far as opinions regarding whether coronavirus is a serious threat or not are concerned, respondents who saw the elderly and medical conditions inequality condition (which emphasizes how the pandemic has been especially hard on the elderly and those with medical conditions) reported significantly lower levels of threat perception compared to respondents who saw the equal pandemic condition (coefficient estimate = -0.166, p-value = 0.001, see left panel of Fig 1). Regarding opinions as to whether the priority should be saving lives or saving the economy, respondents who saw the elderly and medical conditions inequality condition reported significantly more support towards saving the economy over saving lives compared to equal pandemic (coefficient estimate = 0.201, p-value = 0.001, see right panel of Fig 1).

Digging deeper into these patterns revealed an interesting treatment-effect heterogeneity. Both of the effects discussed in the previous paragraph are mainly driven by respondents who are neither at risk themselves nor have family members who are at risk. Significant treatment

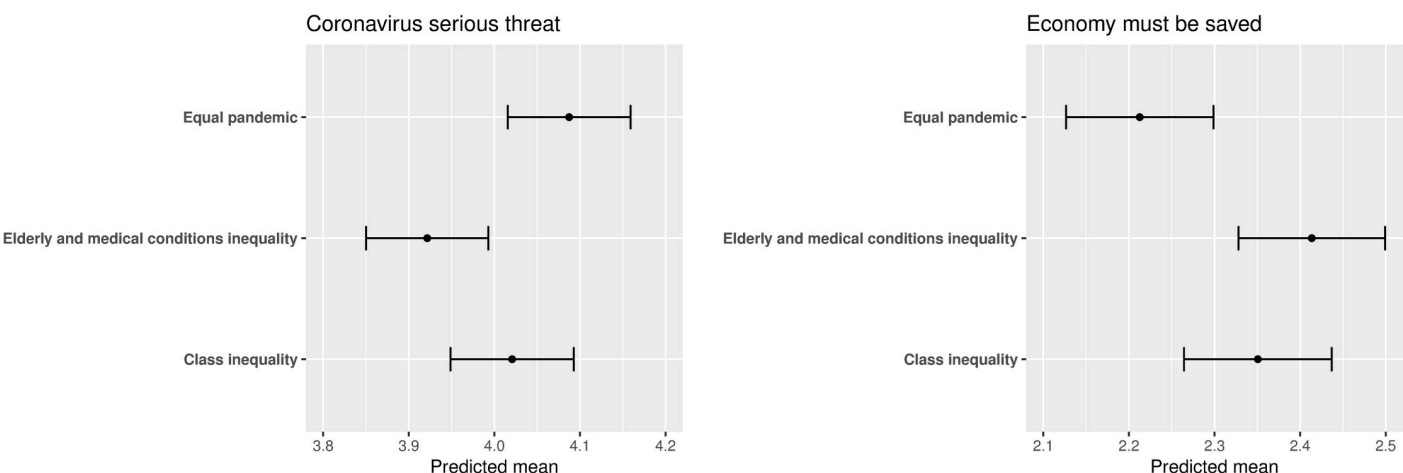

**Fig 1. The effect of the informational treatment on outcomes.** The point estimates are predicted means. The bars denote 95% confidence intervals. *N* = 2,617.

effects are observed only in this not-at-risk group, while the treatment effect decreases in magnitude by more than half and loses statistical significance among respondents who are at risk or have at risk family members (see Fig 2). Effect heterogeneity is demonstrated by fitting separate models for at-risk and not-at-risk sub-groups (i.e., fitting two separate regressions of the outcome on the experimental conditions, one on the at-risk sample and the other on the not-at-risk sample). The elderly and medical conditions inequality coefficient estimate for the outcome "coronavirus serious threat" is -0.061 (p-value = 0.454) for respondents at risk and -0.184 (p-value = 0.004) for respondents not at risk. Similarly, the elderly and medical conditions inequality coefficient estimate for the outcome "economy must be saved" is 0.101 (p-value = 0.348) for respondents at risk and 0.226 (p-value = 0.003) for respondents not at risk.

In addition to the procedure described here to investigate effect heterogeneity, the researcher fitted additional, pooled models that explicitly modeled the outcome as a function of the experimental conditions, the at-risk variable, and interactions between the two. The interactions from these models are insignificant, which is not surprising because the experiment was not powered to be able to detect interaction effects. That said, the at-risk main effects are significant and both the at-risk main effects and interactions are in the expected direction (i.e., opposite of the treatment effects), which explains why the treatment effects are drastically smaller—two to three times—in the at-risk sub-sample. (See S3 Appendix for results based on the models with interactions.)

While the elderly and medical conditions inequality condition led to significant changes in both outcomes, the class inequality condition was weaker in its effects. Despite the effect being in the same direction as elderly and medical conditions inequality, class inequality led to significant changes only in the "economy must be saved" outcome. The class inequality coefficient estimates are -0.067 (p-value = 0.199) for the "coronavirus serious threat" outcome, which is less than half the magnitude of the elderly and medical conditions inequality effect, and 0.138 (p-value = 0.027) for the "economy must be saved" outcome, which is about only two-thirds of the elderly and medical conditions inequality effect. See Table 1 for a compact presentation of the estimated coefficients associated with the experimental conditions for both outcomes. (The statistically significant class inequality effect disappears when we control for the socio-demographic covariates.) On the other hand, data show that the class inequality condition had a nearly significant negative effect of -0.111 (p-value = 0.058) on satisfaction with state's handling of the coronavirus situation; no significant effects are observed for elderly and

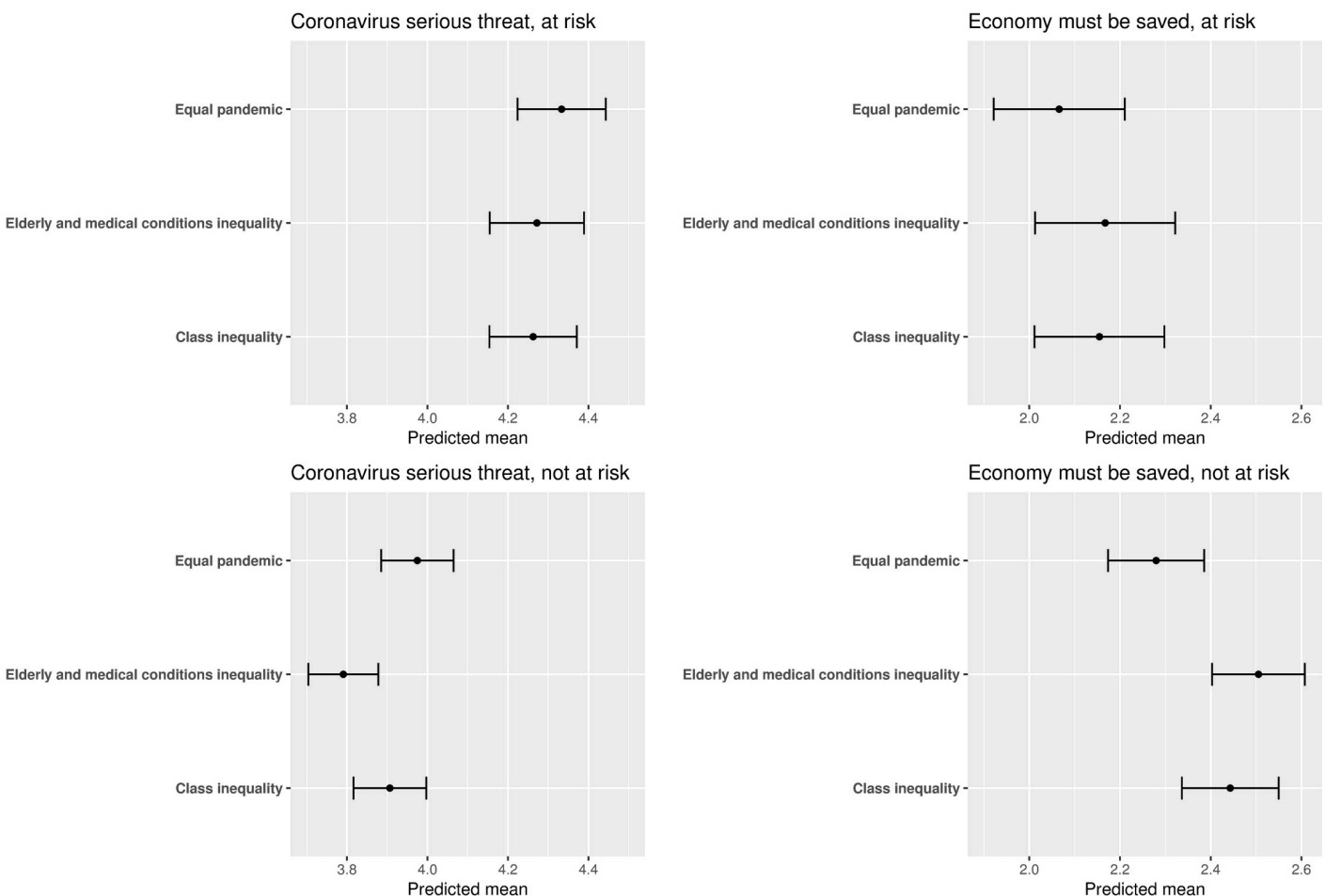

**Fig 2. Effect heterogeneity based on being at risk.** The point estimates are predicted means. The bars denote 95% confidence intervals. $N = 2,617$.

medical conditions inequality or for the other two satisfaction outcomes (city and federal government).

## Discussion

The information the public receives regarding the coronavirus outbreak influences their threat perceptions and whether they think saving the economy or saving lives should be the priority.

**Table 1. Treatment effect estimates.**

|                                              | Coronavirus serious threat | Economy must be saved |
| -------------------------------------------- | -------------------------- | --------------------- |
| Elderly and medical conditions inequality    | -0.166 (0.052)**           | 0.201 (0.062)**       |
| Class inequality                             | -0.067 (0.052)             | 0.138 (0.062)*        |

The numbers inside the parentheses are standard errors. Estimates are based on models without any demographic covariates. Stars denote p-values: ʾp<0.1

* p<0.05

** p<0.01

*** p<0.001.

Results from this study show that being informed about the disproportionate negative impact of the pandemic on the elderly and those with underlying medical conditions make people less likely to see coronavirus as a threat and more likely to prioritize saving the economy as opposed to saving lives, particularly among those who do not need to worry about themselves or someone in their family being at risk of severe illness.

These findings suggest that the dissemination of scientific information regarding the unequal impact of the pandemic on certain groups could actually be causing the general public to become less concerned about the outbreak and its human toll. The fact that the effect is primarily observed among people not at risk further indicate that when those people are sensitized to the situation of the weak they feel more secure about their own situation as not being at risk, which likely leads to increased optimism bias [18] and underestimation of their risk of infection [19]. These results give more support to mechanisms of deliberation and callousness as opposed to sympathy [20–22].

While information regarding the disproportionate negative impact of the pandemic on the elderly and those with underlying medical conditions had a significant impact on coronavirus threat perceptions and preferences regarding whether saving lives or saving the economy should be the priority, information regarding the disproportionate negative impact of the pandemic on the poor did not have as big of an impact on the outcomes and generally failed to achieve statistical significance. One possible explanation for this null effect is that issues around class are highly politicized in the US, and so it is more difficult to move people's opinions on these topics compared to a more neutral and directly health-related topic such as the elderly and those with medical conditions.

The findings also have important policy implications. If the policy goal is to increase caution among the general public and make them take the situation more seriously, then information that emphasizes solidarity—"we are all in this together"—is likely to be much more effective, especially when it comes from a credible source [23, 24]. This solidarity framework should be employed even when informing the public about the unequal impact of the pandemic on certain groups, so that the general public is not left with the impression that the outbreak concerns only some—not all—of us.

## Limitations

One of the limitations of the study is that the 'elderly and medical conditions inequality' and 'class inequality' conditions are completely separate from one another by design. This is justified because the study is primarily concerned with how people understand the impact of the pandemic, not about the actual facts. That said, it is certainly the case that the poor are more likely to have medical conditions as a matter of science, and the current study does not look at this issue that concerns how 'elderly and medical conditions inequality' and 'class inequality' angles intersect. Another, related limitation is that the experimental text used for the 'class inequality' condition mentions minorities when discussing the impact of the pandemic on the poor and low-income communities, which means that the condition refers to not only class but also race disadvantage. Once again, while this choice is justified by virtue of the fact that the framing is in line with the usual way the topic is discussed in public discourse—see, e.g., the recent United States Joint Economic Committee report on coronavirus [25]—the literature on group cues [26] tells us that whether the information is interpreted primarily in terms of class or race will likely influence the way respondents answer survey questions. Therefore, investigating how class and race axes intersect would be a fruitful area of future work. A final limitation is that the custom script used for parsing the manipulation check question is developed by the researcher alone and was not independently verified prior to data collection.

## Supporting information

**S1 Appendix. Experimental texts, images, videos, and other related content.**
(PDF)

**S2 Appendix. Sample size calculations and sample characteristics.**
(PDF)

**S3 Appendix. Regression results.**
(PDF)

**S1 File.**
(ZIP)

**S1 Data.**
(ZIP)

## Acknowledgments

The author thanks David Harding, Dennis Feehan, Daniel Schneider, Gabriel Lenz, Don Moore, and Xinyi Zhang for all of their helpful comments and Anam Ahmed for kindly taking the time to record the experimental texts used in the study.

## Author Contributions

**Conceptualization:** Ugur Yildirim.

**Funding acquisition:** Ugur Yildirim.

**Investigation:** Ugur Yildirim.

**Methodology:** Ugur Yildirim.

**Visualization:** Ugur Yildirim.

**Writing – original draft:** Ugur Yildirim.

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
