## [Decision Letter · Decision Letter 0]

13 Aug 2020

PONE-D-20-17174

Disparate Impact Pandemic Framing Decreases Public Concern For Health Consequences

PLOS ONE

Dear Dr. Yildirim,

Thank you for submitting your manuscript to PLOS ONE. After careful consideration, we feel that it has merit but does not fully meet PLOS ONE’s publication criteria as it currently stands. Therefore, we invite you to submit a revised version of the manuscript that addresses the points raised during the review process.

We look forward to receiving your revised manuscript.

Kind regards,

Jim P Stimpson, PhD

Academic Editor

PLOS ONE

Additional Editor Comments:

The reviewers carefully reviewed this paper and made several excellent suggestions for improvement. The author should respond to the analytical issues raised by reviewers including the choice of linear regression and the choice to have an interaction effect with treatment group. How were class inequality or race included in the inequality frame and randomization? If not included, why not and provide a discussion and include limitations? If so, make this much clearer and include results and discussion. Related, please clarify the frames, particularly “natural inequality.” There should be a more robust description of methodological limitations. Finally, improve the presentation of results so that the paper is easier to read (e.g. first outcome and second outcome language) and the tables should stand on their own without reference to the text.

Journal Requirements:

Reviewers' comments:

Reviewer's Responses to Questions

**Comments to the Author**

1. Is the manuscript technically sound, and do the data support the conclusions?

Reviewer #1: Yes

Reviewer #2: Partly

Reviewer #3: Yes

Reviewer #4: Yes

2. Has the statistical analysis been performed appropriately and rigorously? 

Reviewer #1: Yes

Reviewer #2: Yes

Reviewer #3: Yes

Reviewer #4: Yes

3. Have the authors made all data underlying the findings in their manuscript fully available?

Reviewer #1: Yes

Reviewer #2: Yes

Reviewer #3: Yes

Reviewer #4: Yes

4. Is the manuscript presented in an intelligible fashion and written in standard English?

Reviewer #1: Yes

Reviewer #2: Yes

Reviewer #3: Yes

Reviewer #4: Yes

5. Review Comments to the Author

Reviewer #1: There are parts where there is some confusion:

- For instance, when I was reading I thought it would be good to remind the reader of what "natural inequality" means when discussing the findings (term mentioned on page 4 but not again until the end of page 7).

- Page 8 “main findings”—first paragraph is how the variables were labeled/defined, why is this in the findings section and not in the methods section?

- Also on page 8—I would mention what “manipulation check” is for readers who may not know exactly what this means.

- Page 9, second paragraph, I think it is informal when using “the first outcome”—I suggest that the outcome be defined and explicitly stated and interpreted to make sure the reader does not have to scroll back up to another section of the paper to know what that outcome is and what it means—this makes the paragraphs in the main findings section hard to follow--I would suggest being much more explicit throughout.

- Where is the rational for why subjects were randomized the way that they were (end of page 4)? Why would elderly and those with medical conditions be one condition? We know that low income and poor populations are more likely to have medical conditions (diabetes, cardiovascular disease, obesity, COPD) that arises from systemic oppression and racism, so why separate poor and low income from medical conditions when they are related as much as being elderly can be related with medical conditions? I would definitely add text on how randomization occurred—decision process and theory of why/how these groups were created. Also, if there is a way to perform some sensitivity analyses which could show us what the results would look like if one group was just elderly, and the other group was low-income and with medical conditions, I would highly suggest including this—at the very least, provide some discussion on this in the manuscript.

Reviewer #2: In results you provide a robust description of the data on the use of the class inequality and also the use of the natural inequality. In the discussion there is less or no explanation of the data in results on the class inequality. Please discuss the class inequality data in discussion in some way.

Please explain how the custom script was developed and whether it was validated in some way by having another person read it or whether it was pre-tested on a sample before its use.

There is not section on limitations. The development of the custom script might be a limitation.

Also can you go to greater lengths in the discussion to clarify second and first outcomes as described in this section below- just make it clearer rather then simply saying the first outcome or the second outcome. I had to keep checking back to remind myself of these outcomes.

While the natural inequality condition led to significant changes in both outcomes, class

inequality condition was weaker in its effects. Despite the effect being in the same direction as

natural inequality, class inequality led to significant changes only in the second outcome (second outcome was what again?)

so were people who had the class inequity frame more or less likely to feel threatened by the virus and how did they feel about saving the economy?

Reviewer #3: This study broadly assesses the impact of message framing on the public's perceptions of threat of COVID-19, and their sensitivity to the outbreak’s human toll. The study obtains data from a large nationally representative sample accessed via a private marketing firm. The study is very interesting, timely and has policy-relevant implications.

My comments are discussed below:

Methods

The authors noted the use of linear regression modeling. However, their outcome variables are measured on a Likert Scale and as such are strictly ordinal variables. Did they test for violations of OLS assumptions? Perhaps they can consider ordinal logistic regression (especially if the assumptions of OLS are violated).

Discussion

One of the more interesting findings of this study is that a focus on the disproportionate negative impact of the virus on the elderly (and NOT on poor and low-income communities) minimizes people's perception of the threat of the virus. Why is this so? While the authors discuss the "elderly finding", the finding (or the lack of finding) for the poor and low-income arm is not discussed at all. This is an important omission that needs to be addressed.

Reviewer #4: This is a clearly written manuscript presenting a clear study asking how framing the COVID-19 pandemic in terms of disparate groups affected influences public response. While there is much to admire about this simple yet important question and design, there are a few places the paper can be improved.

The author should be much clearer about whether the treatment group is manipulating class inequality or race in the inequality frame. I pulled the supplementary materials to find the text of this manipulation and it is definitely a conflation of class and race (“poor and low-income communities, particularly minorities such as blacks and Hispanics, have been disproportionately affected”). The paper starts by discussing this as just class inequality, but occasionally also discusses race. There is a very broad literature in social science on group-cues (see e.g., Nelson and Kinder 1996) that is not discussed at all. This is curious, as sociology, political science, and communication all have spent much attention on looking at how framing information in terms of the groups affected – particularly as it relates to class and race – matters. More attention to framing and race/class would be warranted in this paper.

Neither the background nor the methods section motivates or pre-specifies the treatment effect heterogeneity findings – among those not at risk and with no family at risk. Since the authors present these results, they should explain why they sought to look at the data by these strata earlier in the manuscript.

Did the authors pre-register their hypotheses and study design?

You report separate models stratified by risk as noted above, but it would also be statistically sound to estimate an interaction term by at-risk vs. not-at risk and the treatment groups. The way the results by subgroup are reported on p. 9 is confusing -- it sounds like the authors are reporting the coefficients on the risk variable and not the treatment groups? I recommend revising the discussion of the treatment effect heterogeneity section so it is more clear and also reports the results of the models estimated with interaction terms.

Authors describe the “equal pandemic” frame as the control, but I would argue that most news coverage emphasizes groups at risk rather than the idea of “equal pandemic”. By using the “equal pandemic” as the omitted/reference category they assume in their interpretation that the inequality treatments are what are driving the results – but given there is no “no media exposure” control group, it could be that the “equal pandemic” is actually the distinct frame. Authors might want to comment on this in the discussion. Looking at the figures, for instance, I would actually not use the term “control” but instead call this condition “Equal pandemic”.

Reference:

Nelson, T. E., & Kinder, D. R. (1996). Issue frames and group-centrism in American public opinion. The Journal of Politics, 58(4), 1055-1078.

6. PLOS authors have the option to publish the peer review history of their article (what does this mean?). If published, this will include your full peer review and any attached files.

Reviewer #1: No

Reviewer #2: **Yes: **Lauri Andress, Ph.D.

Reviewer #3: No

Reviewer #4: No

---

## [Author Response · Author response to Decision Letter 0]

22 Sep 2020

We thank the editor and the reviewers for their constructive and helpful comments. We carefully reviewed the points raised by all four reviewers and revised the manuscript accordingly. Our responses to specific points raised are provided in the attached Response to Reviewers document.

---

## [Decision Letter · Decision Letter 1]

13 Oct 2020

PONE-D-20-17174R1

Disparate impact pandemic framing decreases public concern for health consequences

PLOS ONE

Dear Dr. Yildirim,

Thank you for submitting your manuscript to PLOS ONE. After careful consideration, we feel that it has merit but does not fully meet PLOS ONE’s publication criteria as it currently stands. Therefore, we invite you to submit a revised version of the manuscript that addresses the points raised during the review process.

Please provide a point-by-point response to the Editor Comments below. You may, if you wish, address other reviewer comments in your revision.

We look forward to receiving your revised manuscript.

Kind regards,

Jim P Stimpson, PhD

Academic Editor

PLOS ONE

Additional Editor Comments (if provided):

Please change “natural inequality” to a more descriptive term representing that construct such as “elderly and medical conditions inequality” in the text and tables/figures.

There is still confusion about whether racial/ethnic minorities are a focal group or not. It’s clear the data exist to analyze this population from the survey instrument but the analysis seems to combine this population with low income persons. If racial/ethnic minorities are not a focus and not genuinely analyzed, then it might be better to drop this reference in the text and tables/figures and instead list this as a limitation and an area of future work.

The paper would benefit from revising the language in the first paragraph of the Experimental Design, especially clearly defining the three groups. If the “equal pandemic” is indeed the control in the experiment, then a more detailed explanation of that group is needed.

Please add a table 2 x 2 perhaps with the treatment on one side of the table and outcomes across the top as suggested by reviewer 2.

Reviewers' comments:

Reviewer's Responses to Questions

**Comments to the Author**

1. If the authors have adequately addressed your comments raised in a previous round of review and you feel that this manuscript is now acceptable for publication, you may indicate that here to bypass the “Comments to the Author” section, enter your conflict of interest statement in the “Confidential to Editor” section, and submit your "Accept" recommendation.

Reviewer #2: (No Response)

Reviewer #4: (No Response)

2. Is the manuscript technically sound, and do the data support the conclusions?

Reviewer #2: Yes

Reviewer #4: No

3. Has the statistical analysis been performed appropriately and rigorously? 

Reviewer #2: Yes

Reviewer #4: Yes

4. Have the authors made all data underlying the findings in their manuscript fully available?

Reviewer #2: Yes

Reviewer #4: Yes

5. Is the manuscript presented in an intelligible fashion and written in standard English?

Reviewer #2: Yes

Reviewer #4: Yes

6. Review Comments to the Author

Reviewer #2: Please add a table 2 x 2 perhaps with the treatment on one side of the table and outcomes across the top

Reviewer #4: The abstract still only mentions disparities on the elderly, those with underlying medical conditions, and the poor and does not mention the disproportionate impact on people of color. This is a part of the study and should be in the abstract.

The literature review still lacks any reference to work framing inequality and/or the effects of group frames / cues on public opinion. The study does not seems situated in to the social scientific context around public understanding of social inequality.

Can you please explain how written consent would have been obtained in an online survey? Did they have to sign something? Most online surveys do not use written consent, in my experience.

What information was provided to respondents about how to answer whether "the respondent or someone in the respondent's family is at risk"? Is this the respondent's own judgment, or based on some kind of medical criteria? While I understand that people's response to the experimental message emphasizing risk to the elderly and those with medical conditions or poor/ POC may be more or less salient based on the respondent's own risk status, it's hard for me to evaluate the value of this stratification until I understand what participants' were referring to when evaluating their own risk. How many people are in each group (at risk vs. not at risk?) I couldn't find this important piece of information. Is the at-risk group simply much smaller than the not-at-risk group (presuming Lucid samples might not include many elderly people who may lack Internet access)? This would explain why results are stronger in the not-at-risk group?

I don't understand why a study design would be pre-registered but not these specific hypotheses - my understanding is that the point of pre-registering is to ensure that the hypotheses tested were a priori and not developed based on what the data actually show.

Language in response to my concern about reference groups (the fact that there is no real control group in this study) is informal - "see the story from the opposite angle". Please clarify for readers.

Please clarify what is meant by "custom script" in the discussion of manipulation check. This is in the response to another reviewer but is not in the manuscript text.

Could the reason that the elderly/medical condition frame reduces threat perception simply because this frame gave people more knowledge about who is at risk? Then, the effect wouldn't be really one about groups vs. universalizing, but about providing specific risk information. This would challenge the premise of the study.

In response to my comment about the fact that the equal pandemic condition is not really a control, the author indicated they no longer refer to the condition as control. However, on p. 11 they say "lower levels of threat perception compared to respondents who saw the control" (twice)

Overall, I'm just not sure that the data support the author's conclusion that dissemination about the unequal impact of the pandemic leads to these outcomes -- I'm not convinced the treatment is about inequality at all, but about discrete risk information. Yes, the two are aligned, but the paper is framed as being about inequality in COVID and not as a risk-information treatment. The class/race treatment not having much of an effect contributes to this conclusion as well. I am not sure I understand the argument as why the class/race manipulation didn't move opinion was because it was politicized. I am not in favor of the stratified analysis by risk, but it seems that a parallel exploratory analysis would divide by either class or race to assess whether there are heterogeneous effects for the other treatment by these salient characteristics. Either way, the lack of pre-specification of these hypotheses -- and scarce contextualization of this study into a body of literature or theory -- leaves me concerned about what this study is trying to accomplish and whether the analysis has achieved it.

7. PLOS authors have the option to publish the peer review history of their article (what does this mean?). If published, this will include your full peer review and any attached files.

Reviewer #2: **Yes: **Lauri A Andress

Reviewer #4: No

---

## [Author Response · Author response to Decision Letter 1]

21 Nov 2020

I thank the editor and the reviewers for their constructive and helpful comments. As suggested by the editor, I carefully reviewed the points raised in the Editor Comments section and revised the manuscript accordingly. Please see the attached response to reviewers document for my responses.

---

## [Editor Report · Decision Letter 2]

25 Nov 2020

Disparate impact pandemic framing decreases public concern for health consequences

PONE-D-20-17174R2

Dear Dr. Yildirim,

We’re pleased to inform you that your manuscript has been judged scientifically suitable for publication and will be formally accepted for publication once it meets all outstanding technical requirements.

Kind regards,

Jim P Stimpson, PhD

Academic Editor

PLOS ONE

---

## [Editor Report · Acceptance letter]

3 Dec 2020

PONE-D-20-17174R2 

Disparate impact pandemic framing decreases public concern for health consequences 

Dear Dr. Yildirim:

I'm pleased to inform you that your manuscript has been deemed suitable for publication in PLOS ONE. Congratulations! Your manuscript is now with our production department. 

Kind regards, 

on behalf of

Dr. Jim P Stimpson 

Academic Editor

PLOS ONE